# Investigation of Phase Shifts Using AUC Diagrams: Application to Differential Diagnosis of Parkinson’s Disease and Essential Tremor

**DOI:** 10.3390/s23031531

**Published:** 2023-01-30

**Authors:** Olga S. Sushkova, Alexei A. Morozov, Ivan A. Kershner, Margarita N. Khokhlova, Alexandra V. Gabova, Alexei V. Karabanov, Larisa A. Chigaleichick, Sergei N. Illarioshkin

**Affiliations:** 1Kotel’nikov Institute of Radio Engineering and Electronics of RAS, Mokhovaya 11-7, 125009 Moscow, Russia; 2Institute of Higher Nervous Activity and Neurophysiology of RAS, Butlerova 5A, 117485 Moscow, Russia; 3FSBI “Research Center of Neurology”, Volokolamskoe Shosse 80, 125367 Moscow, Russia

**Keywords:** phase shift, electromyogram, EMG patterns, Hilbert transform, cross-wavelet spectrum, cross-wave trains, AUC diagrams, differential diagnosis, Parkinson’s disease, essential tremor

## Abstract

This study was motivated by the well-known problem of the differential diagnosis of Parkinson’s disease and essential tremor using the phase shift between the tremor signals in the antagonist muscles of patients. Different phase shifts are typical for different diseases; however, it remains unclear how this parameter can be used for clinical diagnosis. Neurophysiological papers have reported different estimations of the accuracy of this parameter, which varies from insufficient to 100%. To address this issue, we developed special types of area under the ROC curve (AUC) diagrams and used them to analyze the phase shift. Different phase estimations, including the Hilbert instantaneous phase and the cross-wavelet spectrum mean phase, were applied. The results of the investigation of the clinical data revealed several regularities with opposite directions in the phase shift of the electromyographic signals in patients with Parkinson’s disease and essential tremor. The detected regularities provide insights into the contradictory results reported in the literature. Moreover, the developed AUC diagrams show the potential for the investigation of neurodegenerative diseases related to the hyperkinetic movements of the extremities and the creation of high-accuracy methods of clinical diagnosis.

## 1. Introduction

The objective of this study was to develop a mathematical background of a differential diagnosis of neurodegenerative diseases based on the analysis of the phase difference in biomedical signals. This paper describes a new mathematical method for analyzing the phase differences in signals. We developed this method for the investigation of electromyographic (EMG) signals in antagonist muscles in the diagnosis of certain neurodegenerative diseases, such as Parkinson’s disease (PD) and essential tremor (ET). The idea of using EMG of antagonist muscles for the differential diagnosis of neurodegenerative diseases has been actively discussed since the 1960s of the last century. However, no generally accepted consensus has been reached about the possibility of the clinical application of this approach to differential diagnosis. Researchers have expressed different and sometimes directly opposite opinions on this issue. In particular, several publications indicate the possibility of the differential diagnosis of PD and ET by analyzing the phase difference in antagonist muscle EMG signals with high accuracy [1,2,3,4,5,6,7,8,9,10,11,12]. However, other authors have discussed about the inapplicability of this approach to the differential diagnosis of PD and ET because it cannot provide acceptable accuracy [13,14,15,16,17,18].

Let us consider the problem of differential diagnosis of PD and ET in more detail. Currently, electromyography is considered one of the most important diagnostic tools for neurodegenerative diseases [19,20,21,22,23,24,25,26,27,28,29]. EMG signals recorded from the antagonist muscles of the patient extremities are of particular interest. The antagonist muscles are complementary muscles, the contraction of which leads to the movement of the limb in opposite directions (flexion vs. extension). Researchers and clinicians distinguish between the synchronous and alternating patterns of antagonist muscle tremor. A synchronous tremor is characterized by the oscillations of the antagonist muscles roughly coinciding in phase (see the example in Figure 1, left). An alternating tremor is characterized by the oscillations of the antagonist muscles being in the opposite phase (see the example in Figure 1, right). An intermediate pattern of tremor is also considered. However, the relationship between the pattern of tremor and the neurodegenerative disease is ambiguous. The widespread opinion among clinicians is that PD is characterized by the alternating tremor pattern, and ET is characterized by the synchronous tremor pattern. In practice, patients with PD and ET may exhibit different tremor patterns. Moreover, tremor may be not observed in a patient during the examination at all. All of this complicates the differential diagnosis of neurodegenerative diseases using EMG signals. Currently, the applicability of the phase analysis of EMG signals for the differential diagnosis of PD and ET remains an open problem.

Phase analysis is widely used in biomedical research [30,31,32,33,34,35,36,37,38,39,40,41,42,43,44]. The Hilbert transform and complex wavelets are usually used for computing the instantaneous phase [45]. The mathematical methods for instantaneous phase visualization, such as phase and frequency spectrograms, have been developed [46]. However, the existing methods of phase analysis are mainly aimed at detecting the synchronization of biomedical signals and not at using the phase as a diagnostic feature of the disease. In particular, the existing mathematical methods of phase analysis do not allow for solving the following problems:The problem of searching frequency ranges. The instantaneous phase primarily depends on the selected frequency band in the spectrum of the signal. Usually, the frequency band is chosen by a neurophysiologist based on existing ideas about the spectral properties of the biomedical signal. However, the phase characteristics of the signal can demonstrate regularities that are unknown in advance. Thus, one of the main tasks that arises when studying the phase of the signals is identifying the frequency ranges where the instantaneous phase exhibits notable neurophysiologic regularities.The problem of searching for a compromise between the phase and amplitude analysis. One of the main advantages of the phase methods is the ability to identify regularities that do not depend on the amplitude of the signal. However, this property of the phase methods can become a disadvantage if the signal contains noise, which is typical for biomedical signals such as EMG. Thus, a compromise must be found between the influence of the phase and the amplitude of the signal on the target neurophysiologic regularities.

We developed special area under the ROC curve (AUC) diagrams to solve the problem of searching for EMG frequency ranges in which notable neurophysiologic regularities are observed (the idea of the AUC diagrams was described in [47]). We discussed two new types of AUC diagrams. The first type is based on the Hilbert transform, which enables us to find frequency ranges where differences exist in the values of the instantaneous phase of EMG signal envelopes for a group of patients with different neurodegenerative diseases. The second type of AUC diagram is based on the cross-wavelet spectra of the signals, which enables us to find frequency ranges in which phase regularities are observed, considering the largest contribution of EMG fragments with high amplitude.

We developed special metrics describing the relationship between paired EMG signals by analogy with the previously developed metrics of the wave train electrical activity [47]. The concept of cross-wave train is introduced. The cross-wave train is a time- and frequency-localized increase in the power spectral density (PSD) of the cross-wavelet spectrum. The cross-wave train is characterized by the central (leading) frequency, maximum PSD, duration in seconds and periods, bandwidth, and instantaneous phase.

The results of our investigation of clinical data using the developed AUC diagrams revealed new neurophysiologic regularities in the EMG signals of the antagonist muscles of patients with PD and ET. Some of the discovered regularities have opposite directions, which allowed us to explain the contradictory results described in the neurophysiologic literature and to provide a foundation for PD and ET discriminative analysis.

Section 2 describes the investigated data set and the data collection method. Section 3 describes a new type of AUC diagram based on the Hilbert transform. Section 4 describes a new type of AUC diagram based on cross-wavelet spectra. The results of the group data analysis and scatter plots are presented in Section 5. A discussion of the results of the data analysis is outlined in Section 6.

## 2. Experimental Setting and Data Acquisition

All of the methods of data analysis described in this paper are illustrated by examples of experimental data processing. The subject of our study was the electromyographic signals in PD patients at the first stage of the disease according to the classical Hoehn–Yahr scale and ET patients. We included patients who had not previously received specific medical treatment or had not taken medication for 1–2 days before the examination. Note that during the first stage of PD, patients develop hyperkinetic movements on only one side of the body. This fact is the main feature that differentiates the first stage of PD from the next stages of the disease, when the second side also trembles. In this study, we considered the EMG signals only on this side of the body with hyperkinetic movements. Thus, we divided the group of PD patients into two groups. The first group included patients with hyperkinetic movements of the left body side (10 persons), and the second group included patients with hyperkinetic movements of the right body side (12 persons), for 22 persons in total. The group of ET patients included 13 persons. The average age of the PD patients with hyperkinetic movements of the left body side was 58 years, the minimum age was 42 years, and the maximum age was 69 years. The average age of the PD patients with hyperkinetic movements of the right body side was 55 years, the minimum age was 38 years, and the maximum age was 70 years. The average age of the ET patients was 54 years, the minimum age was 20 years, and the maximum age was 85 years. No statistically significant differences between the ages of these three groups of patients were observed (Mann–Whitney test; the alpha level was 0.05). All of the patients were right-handed. The patients were examined at the FSBI Research Center of Neurology, and they were clinically diagnosed.

During the data acquisition, the persons sat in an armchair in a relaxed state. Their legs touched the floor with the entire sole. Their arms rested on the armrests, and the palms hung down. We fitted the patient with EMG electrodes on both arms on the antagonist muscles of the wrist joint (the flexor muscle was musculus flexor carpi radialis, and the extensor muscle was musculus extensor carpi radialis longus) (see Figure 2). The eyes were closed during the measurements. The recording was performed for 1 min and 30 s. A multifunctional system for neurophysiological research Neuron-Spectrum-5 (Neurosoft Ltd., Ivanovo, Russia) was used to record the EMG signals. The EMG sampling rate was 500 Hz. We used a high-pass filter with a cut-off frequency of 0.5 Hz and a 50 Hz notch filter during the recording.

The preprocessing of EMG signals included the following steps:We applied 50, 100, 150, and 200 Hz notch filters to remove the network interference and its harmonics.We applied an 8th-order Butterworth bandpass filter with a bandwidth from 60 to 240 Hz in the forward and reverse directions to isolate the frequency band of the electrical activity of the muscles.We applied the Hilbert transform to compute the envelope of the EMG signal. This envelope was considered as the tremor signal following the classical method of O. E. Khutorskaya [48,49].We decimated the signal to speed up the computation. The decimation coefficient was 4.

This preprocessing procedure yielded ready-for-analysis tremor signals (see the example in Figure 1). In this study, we developed two different methods for analyzing tremor signals based on AUC diagrams. Next, we describe each method and its benefits.

## 3. A Naïve Approach to Phase Difference Analysis

The simplest and most natural method to determine the phase shift in tremor signals in antagonist muscles is to calculate the instantaneous phases of these signals using the Hilbert transform and subtract one phase from another. The instantaneous phase is calculated in the standard method:(1)Ψ(t)=angle(hilbert(x(t)))
where Ψ(t) is the instantaneous phase of the x(t) signal, *t* is time, hilbert is a function that calculates an analytical addition of the x(t) signal, and angle is a function that calculates the angle of the complex number using a four-quadrant arctangent. Note that the phase is a cyclic but not linear measure. Thus, the rule of vector subtraction is used for the subtraction of phases:(2)$$\Delta \Psi \left(t\right)=angle(exp\left(i\Psi_{1}\left(t\right)\right)-exp\left(i\Psi_{2}\left(t\right)\right))$$
where ΔΨ(t) is the difference of phases $\Psi_{1}\left(t\right)$ and $\Psi_{2}\left(t\right)$. In other words, we convert the phases to the complex form, subtract these complex numbers, and compute the angle of the obtained complex number. Note that the Hilbert transform is applied both in the preprocessing stage (to compute the envelopes of the EMG signals) and for the computation of the instantaneous phase of the tremor signal.

The instantaneous phase depends on the selected frequency band of the investigated signal. We do not know in advance what frequency contains interesting neurophysiological regularities. Thus, we need to investigate the instantaneous phase in different frequency bands. To solve this problem, we developed a special kind of AUC diagram based on the Hilbert transform named the Phase-difference AUC diagram. We relied on a previously described methodology for constructing AUC diagrams [47]. The novelty is that the AUC diagrams are used for the analysis of the average differences of the instantaneous phases in various frequency ranges.

Let us consider an example of the Phase-difference AUC diagram (see Figure 3). This AUC diagram is a kind of Frequency AUC diagram; that is, the AUC diagram indicates the AUC values corresponding to different frequency ranges. The abscissa axis is the lower bound of the frequency range (MinFreq); the ordinate axis is the upper bound of the frequency range (MaxFreq). The diagram has a triangular shape because the upper bound is always larger than the lower bound. The AUC values are displayed using a Jet colormap. Small AUC values are indicated in blue; large AUC values are indicated in red. Intermediate AUC values close to 0.5 are indicated in green. We are looking for AUC values close to 0 or 1. The frequencies from 1 to 50 Hz are considered with a step size of 0.1 Hz.

Figure 3 compares the left arms with hyperkinetic movements in patients with PD and ET. Each cell in the AUC diagram corresponds to a certain frequency range. The signal is filtered by the 8th-order Butterworth bandpass filter to extract the frequency range in the tremor signal. The filter is applied in the forward direction and then in the reverse direction. Then, the instantaneous phase of the signal is computed using Equation (Equation 1). The difference in the instantaneous phases of the flexor and extensor muscles’ tremor is calculated for each patient using Equation (Equation 2). Then, the average value of the instantaneous phase difference is computed for each patient using Equation (Equation 3):(3)$$\mu \Delta \Psi =\frac{1}{N}\sum_{n=1}^{N}\Delta \Psi \left(n\right)$$
where *N* is the number of points in the tremor signal.

The set of average values of the instantaneous phase differences in PD patients is compared with the set of average values of the instantaneous phase differences in ET patients. The comparison is performed using the standard ROC analysis. The area under the ROC curve (AUC) is calculated. The AUC values are displayed on the diagram using a colormap.

Several areas of red and blue are shown in the Figure 3. These areas correspond to the frequency ranges where differences between the instantaneous phase shift in the antagonist muscles of PD and ET patients are observed. The simultaneous presence of the red and blue areas indicates that neurophysiological regularities in opposite directions are observed. Let us consider the largest single-color areas in the AUC diagram.

We can observe a red area in frequency ranges from 0 to 9 Hz along the abscissa axis and from 5 Hz and above along the ordinate axis. The brightest point has coordinates of 5 Hz on the abscissa axis and 11 Hz on the ordinate axis. This point corresponds to the frequency range from 5 to 11 Hz. The red indicates that PD patients have a larger shift in the instantaneous phase in antagonist muscles tremor than ET patients. The AUC value is approximately 0.7 in this frequency range. It makes sense to check whether there is a statistically significant difference between the sets of average values of the instantaneous phase differences in the frequency range from 5 to 11 Hz. The Mann–Whitney test did not reveal a significant difference in this point. However, we observed a statistical tendency (*p*-value < 0.12). We return to this point in the next section: we observed statistically significant differences using another type of AUC diagram.

Another red area is observed in the frequency ranges from 13 to 18 Hz on the abscissa axis and from 20 to 35 Hz on the ordinate axis. The brightest point has coordinates of 15 Hz on the abscissa axis and 23 Hz on the ordinate axis. This point corresponds to the frequency range from 15 to 23 Hz. This colored area is characterized by the instantaneous phase shift being larger in PD patients than in ET patients, just as in the previous case. The AUC value is approximately 0.71 in this frequency range. The results of the Mann–Whitney test did not reveal a significant difference in this point. However, we observed a statistical tendency (*p*-value < 0.11).

A blue area is observed in frequency ranges from 20 to 40 Hz on the abscissa axis and from 25 Hz and above along the ordinate axis. The brightest point has coordinates of 31 Hz on the abscissa axis and 41 Hz on the ordinate axis. This point corresponds to the frequency range from 31 to 41 Hz. Thus, the PD patients have smaller instantaneous phase shifts between antagonist muscles than the ET patients in this frequency range. The AUC value is approximately 0.19 in this frequency range. Note that the blue area consists of several small blue areas. This indicates that we simultaneously observed several neurophysiological regularities of opposite directions in this frequency range. One regularity manifests stronger than the other depending on some unknown factors. The results of the Mann–Whitney test indicated a statistically significant difference in this point (*p*-value < 0.015).

Let us consider a Phase-difference AUC diagram corresponding to the right arms of PD and ET patients (see Figure 4). Red and blue areas are also observed in this diagram. The coordinates of the red area range from 10 to 22 Hz on the abscissa axis and from 20 Hz and above on the ordinate axis. The brightest point has coordinates of 17 Hz on the abscissa axis and 31 Hz on the ordinate axis. The AUC value is approximately 0.83 in this frequency range. The results of the Mann–Whitney test indicated a statistically significant difference in this point (*p*-value < 0.006). Note that this area overlaps the central red area in the AUC diagram of the patients’ left arms (Figure 3). Based on this observation, we hypothesized that patients with hyperkinetic movements in the right arm demonstrate the same neurophysiological regularities as patients with hyperkinetic movements in the left arm. However, these regularities are combined in different proportions. The red area corresponding to the right arm is larger in comparison with that of the left arm, resulting in the blue area being almost invisible.

Thus, we demonstrated that the naïve approach to analyzing the phase difference of the antagonist muscles using Butterworth filters and the Hilbert transform allows one to identify the presence of neurophysiological regularities in opposite directions. Another observation was of the sufficient difference between the groups of patients with hyperkinetic movements in the left and right arms. However, the expressive capabilities of the Phase-difference AUC diagram were insufficient for a detailed investigation of the observed regularities. In particular, we could not investigate other properties of antagonist muscle tremor signals related to the regularities such as PSD, duration, or bandwidth.

A notable disadvantage of the Phase-difference AUC diagrams described in this section is that the instantaneous phases are investigated without considering the amplitude of the signal. EMG signals contain a sufficient noise component, as do other biomedical signals. Noise is characterized by a random change in the instantaneous phase. This makes neurophysiological regularities invisible within the background noise. To develop a more effective approach to analyzing noisy signals, we must find a compromise between the influence of the phase and amplitude components during the analysis of the signal. To solve this problem, we developed an advanced type of AUC diagram, which we describe in the next section.

## 4. AUC Diagrams Based on Cross-Wavelet Spectra

The cross-wavelet spectrum (CWS) is the product of wavelet transforms of a pair of signals *x* and *y*:(4)$$W_{xy}\left(t,s\right)=W_{x}\left(t,s\right)W_{y}^{*}\left(t,s\right)$$
where $W_{x}\left(t,s\right)$ is the wavelet transform of the signal *x*, $W_{y}^{*}\left(t,s\right)$ is the complex conjugation of the wavelet transform of the signal *y*, *t* is time, and *s* is the scaling factor (a value inversely proportional to the frequency).

We use the complex Morlet wavelet for the wavelet transforms of the signals (Equation 5):(5)$$M\left(t\right)=\frac{1}{\sqrt{\pi F_{b}}}exp\left(2\pi ıF_{c}t\right)exp\left(\frac{-t^{2}}{F_{b}}\right)$$
where $F_{b}$ is the bandwidth coefficient, and $F_{c}$ is the central frequency coefficient ($F_{b}$ = 1 and $F_{c}$ = 1).

Cross-wavelet spectra have the following useful properties:The power of the cross-wavelet spectrum $|W_{xy}\left(t,s\right)|$ indicates a degree of similarity of the PSD of signals *x* and *y*.The angle of the complex value of the cross-wavelet spectrum $angle(W_{xy}\left(t,s\right))$ indicates a phase shift between the signals *x* and *y*.

Thus, we can identify the time and frequency points corresponding to a simultaneous increase in the PSDs of signals *x* and *y* by examining the power of the cross-wavelet spectrum of the antagonist muscles. Then, we can investigate the phase shift between the signals at these points by examining the angle of the complex values of the cross-wavelet spectra.

Let us consider an example of the cross-wavelet spectrum in Figure 5. We computed the cross-wavelet spectrum for the tremor signals in the antagonist muscles of the left arm in a first-stage PD patient. The power of the cross-wavelet spectrum is displayed using a Jet colormap. The red and blue correspond to the maximum and minimum values of the spectrum, respectively. The spectrum demonstrates a pronounced maximum at a frequency of 5.3 Hz and at a time of 53.39 s. This maximum corresponds to a simultaneous increase in the power of the wavelet spectra. The value of the phase difference in the specified point is −3.03 radians. The phase shift in the indicated point is more notable than the phase shift in the neighborhood points because we can assume that the signal-to-noise ratio reaches a maximum at this point. The idea for our analysis was to investigate the phase difference in signals at such points.

Figure 6 demonstrates the $angle(W_{xy}\left(t,s\right))$ phase shift of the cross-wavelet spectrum given in Figure 5. The diagram is mainly blue, which indicates the phase shift between the antagonist muscles of approximately −2.68 radians (alternating pattern of tremor). The neighborhood of the maximum of the cross-wavelet spectrum of the PSD is dark blue; the phase shift is approximately −3.03 radians. Thus, we can say that the values of the phase shift are more stable and closer to −π in the local maximum area of the PSD of the cross-wavelet spectrum.

We applied the method developed earlier for the analysis of the wave train electrical activity [42,47,50,51,52,53,54,55,56,57,58,59,60,61,62,63] to analyze the local maxima in the cross-wavelet spectra. The method is based on the use of 2D and 3D AUC diagrams. The AUC diagrams of the cross-wavelet spectra (CWS AUC diagrams) are used analogously to the AUC diagrams developed for the analysis of wave trains [47]. Thus, we can introduce the following kinds of CWS AUC diagrams: Frequency CWS AUC diagrams, PSD CWS AUC diagrams, Duration CWS AUC diagrams, and Bandwidth CWS AUC diagrams. Here, we propose a new type of AUC diagram in addition to the listed AUC diagrams, namely, the Phase CWS AUC diagrams, which we discuss below.

The principle of the construction of and the rules for reading AUC diagrams were the same as previously [47]. However, the physical meaning of the CWS AUC diagrams is somewhat different. The CWS AUC diagrams indicate the central frequency, duration, bandwidth, and other attributes of areas in the cross-wavelet spectrum where an increased similarity between two signals is observed. We call the areas of similarity between the signals cross-wave trains, analogous to wave trains.

Let us create a Frequency CWS AUC diagram for tremor signals in antagonist muscles in the left arm with hyperkinetic movements of PD and ET patients (see Figure 7). The diagram is mainly dark blue. A bright red area is observed only in the frequency range of 4–6 Hz, associated with hyperkinetic movements. Note that the blue corresponds to AUC values close to 0. This indicates that the considered characteristics of the signals may have diagnostic value. An AUC close to 0 indicates that the cross-wavelet spectra of ET patients contain areas of similarity between the tremor signals, which are not typical for PD patients. An AUC close to 1 (the red areas in the diagram) indicates that the cross-wavelet spectra of the PD patients contain areas of similarity that are not typical for ET patients. These AUC values may also have diagnostic value. Thus, the Frequency CWS AUC diagram indicates two regularities with opposite directions in the tremor signals. One of these regularities relates to the frequency range of hyperkinetic movements. The second regularity may relate to the physiological tremor of patients.

The Frequency CWS AUC diagram Figure 7 confirmed the conclusions we had drawn based on the Phase-difference AUC diagrams considered in the previous section. However, the advantage of the CWS AUC diagrams is that they allow us to apply the iterative procedure of fitting the cross-wave train characteristic by analogy with the analysis of wave trains [47].

The analysis of EMG signals of PD and ET patients yielded the following results (Table 1 and Table 2). Table 1 contains the results of the analysis of the left arms of PD and ET patients with hyperkinetic movements of the left arm. The iterative procedure of fitting the characteristics of cross-wave trains revealed at least four kinds of cross-wave trains that allowed one to distinguish the PD and ET patients. These four kinds of cross-wave trains differ in the central frequency and other parameters. Two of the four kinds of cross-wave trains are characterized by AUC values close to 1 (labeled as the first and second red areas). Following the semantics of AUC diagrams, AUC values close to 1 indicate that cross-wave trains of given kinds are typical for PD patients but not for ET patients. Two other kinds of cross-wave trains are characterized by AUC values close to 0 (labeled as the first and second blue areas). The cross-wave trains of these kinds are typical for ET patients but not for PD patients. The results of the Mann–Whitney test confirmed that the number of cross-wave trains significantly differs in PD and ET patients (see the *p*-values in the last column of Table 1 and Table 2). We confirmed this finding for all kinds of cross-wave trains.

Let us consider in detail the characteristics of cross-wave trains that are typical for PD patients. The instantaneous phase is the most interesting characteristic of the cross-wave trains because it indicates which pattern of tremor, synchronous or alternating, they belong to. We have developed a special kind of CWS AUC diagram, the so-called Phase CWS AUC diagram, for the investigation of the cross-wave train instantaneous phase. The principle of constructing Phase CWS AUC diagrams is the same as that of other kinds of CWS AUC diagrams. The AUC values are calculated using the standard ROC analysis. These values are indicated using a Jet colormap (see the example in Figure 8).

A special feature of Phase CWS AUC diagrams is that they contain two triangular regions and not one region. The upper triangular region has the usual meaning. In this region, we must account for all cross-wave trains that belong to the phase range indicated using the abscissa axis (the lower bound of the range) and the ordinate axis (the upper bound of the range). The lower triangular region has the following meaning: in this region, we account for all cross-wave trains *with the exception* of those that belong to the phase range indicated using the abscissa axis (the *upper* bound of the range) and the ordinate axis (the *lower* bound of the range). This scheme of Phase CWS AUC diagrams is necessary because the phase is a cyclic but not linear measure.

The example in Figure 8 demonstrates a Phase CWS AUC diagram corresponding to cross-wave trains observed in the frequency range from 4 to 6 Hz. This frequency range corresponds to the frequencies of Parkinsonian tremor. Thus, it is not a surprise that we can observe a large yellow area in the center of the upper triangular region. Yellow indicates an AUC close to 0.6. In the lower and upper parts of the upper triangle, on the contrary, we can observe AUC values close to 1 (dark red areas). The considered Phase CWS AUC diagram indicates that the analyzed cross-wave trains have an instantaneous phase close to −π or +π. Thus, these cross-wave trains, as expected, belong to the alternating pattern that is considered typical for PD patients. An example of the cross-wave train in the frequency range from 4 to 6 Hz is demonstrated in Figure 9.

Let us consider the Phase CWS AUC diagram related to the second frequency range of 8–18 Hz (see Figure 10). A bright red area is observed in the center of the upper triangle. It means that the considered cross-wave trains have the instantaneous phase mainly close to 0, that is, the cross-wave train belongs to the synchronous pattern of the tremor. An example of the cross-wave train in the frequency range from 8 to 18 Hz is given in Figure 11.

Thus, the results of our analysis revealed that PD patients with hyperkinetic movements of the left arm demonstrate simultaneously alternating and synchronous tremor patterns. The cross-wave trains corresponding to these tremor patterns differ in amplitude by almost two times. In addition, these cross-wave trains differ in the duration and frequency bandwidth (see Table 1). This explains why researchers have reported contradictory findings about tremor in PD patients.

ET patients are also characterized by at least two kinds of cross-wave trains that distinguish them from PD patients. These cross-wave trains differ in the central frequency, amplitude, duration, and frequency bandwidth (see Table 1). We did not reveal any sufficient inhomogeneity in the instantaneous phase values in the frequency range from 2.2 to 3.9 Hz. The cross-wave trains in the frequency range from 10 to 37 Hz also demonstrate various values of the instantaneous phase. Notably, the upper triangular region of the Phase CWS AUC diagram contains a pale blue area in the center (see Figure 12). This means that the cross-wave trains in the frequency range from 10 to 37 Hz mainly correspond to the alternating pattern of the tremor. This finding is especially notable because the considered cross-wave trains are typical for ET patients but not PD patients. An example of the cross-wave train in the frequency range from 10 to 37 Hz is depicted in Figure 13.

The parameters of cross-wave trains detected on the right arms of PD and ET patients with hyperkinetic movements of the right arm are provided in Table 2. We revealed at least four kinds of cross-wave trains in the right arms of the patients. These kinds of cross-wave trains approximately correspond to the four kinds of cross-wave trains detected in the left arms of patients. The central frequency, amplitude, duration, and frequency bandwidth of cross-wave trains in the left and right arms are slightly different. However, we did not observe inhomogeneity in the instantaneous phase values in the right arm. In other words, we could not confirm that the cross-wave trains in the right arm belong to the synchronous or alternating tremor patterns. This difference between the left and right arms of the patients is yet another reason for the contradictory reports in the neurophysiological literature, because most researchers did not separate PD patients with hyperkinetic movements in the left and right arms. Note that the AUC values in Table 1 and Table 2 are close to 0 or 1. This means that cross-wave trains can be successfully used for the clinical diagnosis of PD patients regardless of which side of the body demonstrates hyperkinetic movements.

**Table 2 sensors-23-01531-t002:** Characteristics of cross-wave trains that distinguish PD and ET patients with right arm hyperkinetic movements.

Investigated Regularity	Frequency, Hz	PSD, μV${}^{2}/$ Hz	Duration, Periods	Bandwidth, Hz	Phase, Rad	AUC	*p*-Value
The first blue area	2.2–3.9	0.01–8	0.6–1.2	0.4–1.9	−π…+π	0.04	0.0002
The first red area	2.5–6	0.2–970	1.6–3	0.5–1	−π…+π	1	0.00002
The second red area	8–12	1.3–860	0.5–1.5	1.5–5	−π…+π	0.93	0.0003
The second blue area	12–37	0.02–24	0.6–0.9	2.6–5.2	−π…+π	0	0.00003

## 5. Statistical Data Analysis

We performed a correlation analysis of the number of cross-wave trains corresponding to different frequency ranges (Table 1 and Table 2).

The scatter plot in Figure 14 (left) shows the number of cross-wave trains per second in the frequency ranges of 4–6 Hz and 8–18 Hz. Red diamonds represent PD patients; yellow diamonds represent ET patients. For both groups of patients, we studied left arms with hyperkinetic movements. The abscissa axis is the number of cross-wave trains per second in the range of 4–6 Hz; the ordinate axis is the number of cross-wave trains per second in the range of 8–18 Hz. The parameters of the cross-wave trains are summarized in Table 1. Each diamond corresponds to one patient. The PD patients’ point cloud is elongated along the abscissa axis. The ET patients’ point cloud is located lower and more left than that of the PD patients. The obtained point clouds are located close to each other but can be easily separated. The results of correlation analysis did not reveal a statistically significant correlation of the number of cross-wave trains in the considered frequency ranges in patients with PD. This allows us to suggest that the considered different kinds of cross-wave trains in PD patients are controlled by different neurophysiological mechanisms. In patients with ET on the left arms, we found a statistically significant correlation between the numbers of cross-wave trains in the considered frequency ranges. The Spearman’s correlation coefficient is 0.73; the probability of the type I error according to the Spearman’s nonparametric test is 0.0039.

The scatter plot in Figure 14 (right) shows the number of cross-wave trains per second in frequency ranges of 2.5–6 Hz and 8–12 Hz obtained for patients’ right arms. The abscissa axis is the number of cross-wave trains per second in the range of 4–6 Hz; the ordinate axis is the number of cross-wave trains per second in the range of 8–12 Hz. The parameters of the cross-wave trains are summarized in Table 2. The locations of the PD and ET right arm point clouds are similar to those of the left arm point clouds. Therefore, the characteristics of cross-wave trains observed in the right and left arms of PD patients are approximately equal. However, in contrast to the patients’ left arms, in ET patients’ right arms, we found no statistically significant correlation between the numbers of cross-wave trains in the considered frequency ranges. At this point, we cannot explain the observed difference in the right and left arms in ET patients. Perhaps this is due to a matter of dominance of the arms.

The scatter plot in Figure 15 (left) shows the number of cross-wave trains per second in frequency ranges of 2.2–3.9 Hz (abscissa axis) and 10–37 Hz (ordinate axis) in PD and ET patients’ left arms with hyperkinetic movements. The scatter plot in Figure 15 (right) shows the number of cross-wave trains per second in frequency ranges of 2.2–3.9 Hz (abscissa axis) and 10–37 Hz (ordinate axis) in patients’ right arms. The cross-wave trains parameters are summarized in Table 1 and Table 2. The PD patients’ point cloud is elongated across a diagonal line. The ET patients’ point cloud is located more on the right and higher than that of the PD patients. The PD and ET point clouds are easy to separate. The results of correlation analysis showed a statistically significant correlation between the number of cross-wave trains in the considered frequency ranges in patients with PD. For the PD patients’ left arms, the correlation coefficient was 0.82, the type I error probability was 0.003, the Spearman’s correlation coefficient was 0.84, and the type I error probability according to the nonparametric Spearman’s test was 0.004. For the PD patients’ right arms, the correlation coefficient was 0.83, the type I error probability was 0.007, the Spearman’s correlation coefficient was 0.82, and the type I error probability according to the nonparametric Spearman’s test was 0.009.

Thus, we found a strong correlation between the occurrence frequencies of cross-wave trains in the frequency ranges under consideration, both on the left and right arms of patients with PD. Therefore, we suggest that both types of cross-wave trains are controlled by the same neurophysiological mechanism. Note that the cross-wave trains in the considered frequency range characterize ET patients and not PD patients. We also found no statistically significant correlation between the occurrence frequencies of the considered types of cross-wave trains in ET patients. This allowed us to suggest that the difference in the cross-wave trains occurrence frequencies in PD and ET patients is due to such cross-wave trains being suppressed in PD patients but is not due to the increase in the number of cross-wave trains in ET patients.

This hypothesis was indirectly confirmed by the results of the analysis of the correlation between the frequency with which cross-wave trains of a different kind appear and patient age. We discovered a statistically significant correlation between patient age and the number of cross-wave trains in the range of 12–37 Hz in the right arms of patients with PD with hyperkinetic movements (see Figure 16, right). The correlation coefficient was −0.56, the type I error probability was 0.05, the Spearman’s correlation coefficient was −0.73, and the type I error probability according to the nonparametric Spearman’s test was 0.009. On the left arms with hyperkinetic movements, we did not observe such regularity (see Figure 16, left). In an earlier study [47], we assumed that age is an indicator of the development of PD. It follows from this assumption that the presence of a negative correlation between the age and frequency of occurrence of physiological tremor cross-wave trains also indicates the existence of a neurophysiological mechanism that suppresses the appearance of such cross-wave trains during the development of PD, as evidenced by the regularity observed in the scatter plots Figure 15.

The results of our study of the dependence of the occurrence frequency of other kinds of cross-wave trains on the age of patients did not reveal the presence of statistically significant correlations.

## 6. Discussion

Clinical doctors distinguish synchronous, alternating, and intermediate patterns of antagonist muscle tremors. Attempts to perform a differential diagnosis of PD and ET, as well as to analyze the response to taking specialized medications by the pattern of tremor, have produced contradictory results [3,5,64,65,66]. Previously, synchronous tremor was thought to be typical for ET, and alternating tremor to be typical for PD [1]. However, at present, this regularity is considered to be related only to the rest tremor [1]. The postural tremor can contain both synchronous and alternating patterns in both of the above pathologies [67,68]. Thus, at present, the predominance of the alternating tremor is not accepted as a reliable clinical sign of PD. We verified below the validity of this hypothesis by a direct experiment using the CWS AUC diagrams.

Let us construct a Frequency CWS AUC diagram using EMG signals in antagonist muscles in the left arm with hyperkinetic movements in PD and ET patients (see Figure 17). Let us constrain the values of the instantaneous phase of cross-wave trains by ranges from −π to −π/2 and from +π/2 to +π. That is, we consider only the cross-wave trains of the alternating tremor. The diagram is mainly dark blue. The red area, related to the hyperkinetic movement frequency range of 4–6 Hz discovered in the Figure 7, became brighter. The AUC in the frequency range of 4–6 Hz is 0.95. Therefore, the doctor can make the correct diagnosis in about 95% cases if s/he tries to use the presence of alternating tremor on the left arm as a hallmark of PD.

Now, let us see what happens if the doctor tries to make the diagnosis by analyzing the tremor in the right arm of the patient. Figure 18 demonstrates the Frequency CWS AUC diagram of EMG in the antagonist muscles of the right arm with hyperkinetic movements in PD and ET patients. The instantaneous phase of cross-wave trains is constrained by the ranges from −π to −π/2 and from +π/2 to +π as in the previous case. No red area appears in the Figure 18, unlike the left-arm Figure 17. The AUC is 0.53 in the frequency range of 4–6 Hz. The probability of making a correct diagnosis is the same as if the doctor tossed a coin. This explains why researchers have published contradictory results on the differential diagnosis of PD and ET if they do not distinguish the left- and right-sided types of PD. Note that some types of cross-wave trains we discussed in this study supplied about 100% differentiation of PD and ET patients. This was achieved because we considered a set of parameters of the cross-wave trains including the duration, PSD, and frequency bandwidth but not only the instantaneous phase.

Currently, neurophysiologists distinguish between physiological and pathological tremors. Physiological tremor usually corresponds to a low-amplitude muscle tremor with a frequency of approximately 8–12 Hz [69,70]. Physiological tremor can be observed in any healthy person. Based on the results of our study, we state that the frequency range of the physiological tremor is much wider and includes at least frequencies from 2.2 to 37 Hz. Moreover, the physiological tremor parameters can serve as a diagnostic feature of various neurodegenerative diseases. Note that we previously reported neurophysiological regularities in lower frequency ranges up to 0.5 Hz [56,59].

Our results indicated the simultaneous presence of two types of cross-wave trains in PD patients with hyperkinetic movements of the left arm. We observed cross-wave trains of the mainly alternating pattern in the frequency range of 4–6 Hz. We observed cross-wave trains of the mainly synchronous pattern in the frequency range of 8–18 Hz. We did not observe such a contradiction in ET patients. To explain this finding, we must consider the mechanisms of hyperkinetic movements in PD and ET patients. The origin of ET is thought to be associated with rhythmic activity in the cortical–pontine–cerebellar–thalamo–cortical loop with output to the spinal cord from the motor cortex. The destruction of the ventral intermediate nucleus (VIM) of the thalamus causes immediate termination of the tremor [71]. The appearance of tremor in PD involves more brain structures. The resting tremor in PD is associated with the cerebellar–thalamo–cortical circuit, the basal ganglia, and the interaction between these two circuits [72]. Some cells fire at a tremor frequency (tremor cells) in the VIM of the thalamus, which receives information from the cerebellum. Tremor cells exist in the subthalamic nucleus (STN) and internal globus pallidus (GPi) [73]. Tremor in PD may occur to compensate for pathological beta oscillations in the motor system [74]. Clinical evidence shows that several nodes within the distributed basal ganglia and the cerebellar–thalamo–cortical network support the tremor pacemaker. Some areas may play a role in triggering the tremor in this distributed network, whereas other areas are involved in maintaining or amplifying the rhythm of the tremor (cerebellar–thalamo–cortical circuit) [75]. Thus, more opportunities exist to trigger various patterns of tremor in PD patients. This issue requires more detailed study.

Our results evidence the presence of fundamental differences in the clinical manifestations and mechanisms of hyperkinetic movements in left- and right-sided PD patients [47]. The explored differences could potentially be generalized with further studies. Usually, researchers do not consider differences between the right and left hemispheres when analyzing the sources of tremor. These studies are yet to be conducted. However, some researchers addressed the role of dominant and nondominant arms in the movements, which have different specializations in implementing movements [76,77]. The dominant arm specializes in controlling the dynamics of smooth and efficient movement [78], whereas the nondominant arm specializes in other functions related to contingency resistance [79,80]. Thus, the dominant and nondominant arms are controlled by different neurophysiological mechanisms, which may interact differently with the mechanisms that produce the tremor in these arms. This may be a reason for the difference in the phase shift of the tremor signals of the antagonist muscles in the dominant and nondominant arms.

## 7. Conclusions

We developed a new method of exploratory data analysis that addressed identifying regularities in paired signals. This method generalizes existing methods of signal synchronization analysis. Our method allows for the identification of frequency ranges where the instantaneous phase of the given signals demonstrates notable neurophysiological regularities. In addition, the method allows us to find a compromise between the influence of the phase and amplitude characteristics of the analyzed signals. We discovered new neurophysiological regularities in the tremor signals of the antagonist muscles in PD and ET patients using the developed method. In particular, we found that the EMG signals of the antagonist muscles simultaneously have several regularities with opposite directions. This finding explains the contradictory results published in the neurophysiological literature. Our study advances the research on neurodegenerative diseases and provides new important information on the mechanisms of tremor appearance and control in PD and ET patients. The discovered regularities can be used to develop high-accuracy methods for the differential diagnosis of PD and ET. In our future research, we will address the development of mathematical metrics differentiating PD and ET patients, which are necessary for clinical applications of neurophysiological regularities reported in this paper. Notably, the developed method is universal and can be applied to the analysis of other types of biomedical signals.

## 8. Patents

Sushkova O.S., Morozov A.A., Gabova A.V., Karabanov A.V. Patent number RU 2741233 C1. Russian Federation. Method for differential diagnosis of essential tremor and early and first stages of Parkinson’s disease using wave train activity analysis of muscles. Published: 22 January 2021 Bul. No. 3. Application: 2020118098, 24 April 2020. Starting date of the patent validity period: 24 April 2020. Date of registration: 22 January 2021. Application date: 24 April 2020. Address for correspondence: 125009, Moscow, st. Mokhovaya, 11/7, Kotelnikov IRE RAS, Patent Department. Accessed date of web is 26 January 2023: https://new.fips.ru/registers-doc-view/fips_servlet?DB=RUPAT&DocNumber=2741233&TypeFile=html.

## Figures and Tables

**Figure 1 sensors-23-01531-f001:**
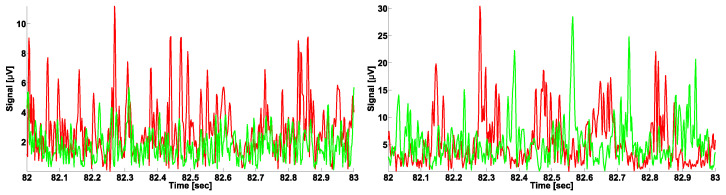
Patterns of tremor. Tremor signals are extracted from electromyographic (EMG) signals. The envelope of the EMG signals is used as the tremor signal; the Hilbert transform is used for computing the envelope of the signal. The signals from the flexor and extensor muscles are indicated in green and red, respectively. (**Left**), the synchronous pattern of tremor. (**Right**), the alternating pattern of tremor. The abscissa is time in seconds. The ordinate is the envelope of the signal in μV.

**Figure 2 sensors-23-01531-f002:**
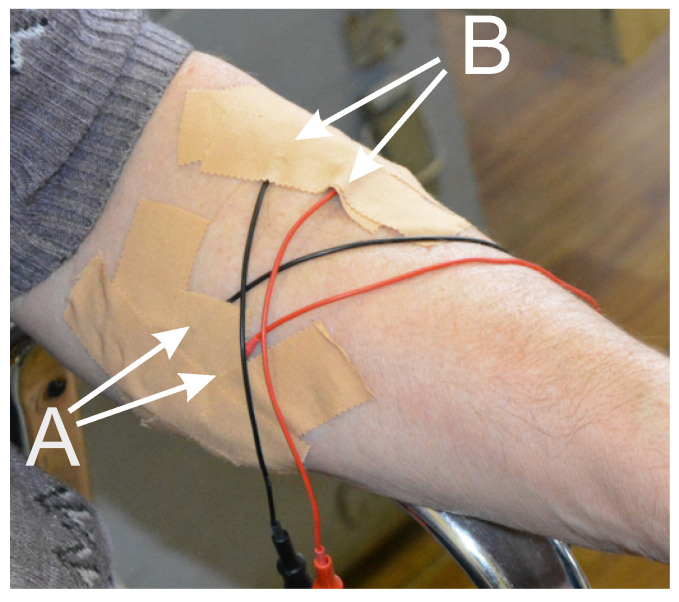
The location of the EMG electrodes on the patient’s arm. A—a bipolar electrode on the flexor muscle musculus flexor carpi radialis. B—a bipolar electrode on the extensor muscle musculus extensor carpi radialis longus.

**Figure 3 sensors-23-01531-f003:**
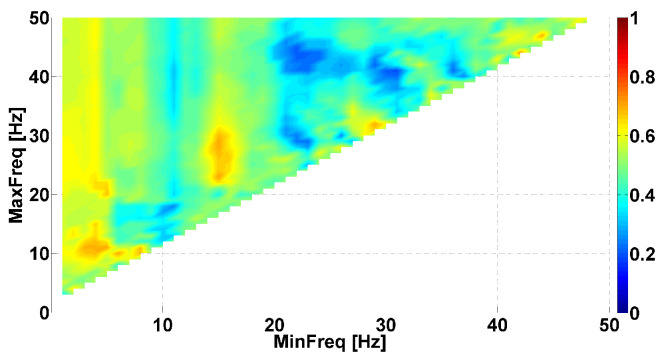
Example of Phase-difference area under the ROC curve (AUC) diagram. This AUC diagram is a kind of Frequency AUC diagram, that is, the AUC diagram indicates the AUC values corresponding to different frequency ranges. Parkinson’s disease (PD) and essential tremor (ET) patients are compared, namely, *left* arms with hyperkinetic movements are analyzed. The abscissa axis is the lower bound of the frequency range; the ordinate axis is the upper bound of the frequency range.

**Figure 4 sensors-23-01531-f004:**
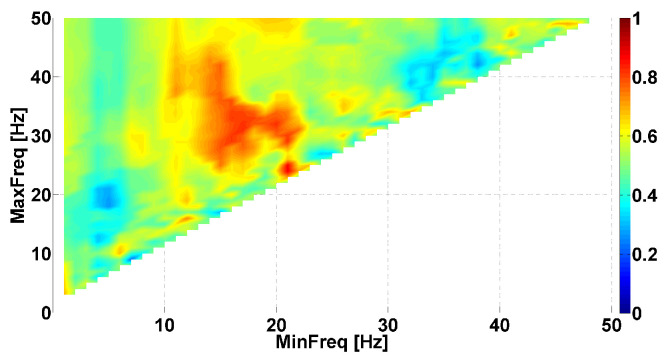
Example of Phase-difference AUC diagram. This AUC diagram is a kind of Frequency AUC diagram. PD and ET patients are compared, namely, *right* arms with hyperkinetic movements are analyzed. The abscissa axis is the lower bound of the frequency range; the ordinate axis is the upper bound of the frequency range.

**Figure 5 sensors-23-01531-f005:**
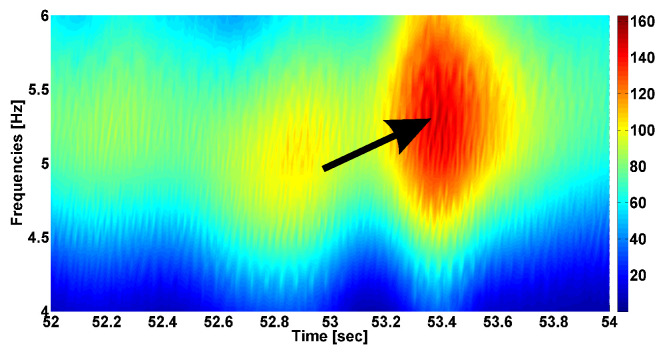
Example of the cross-wavelet spectrum of tremor signals in antagonist muscles in a first-stage PD patient. Signals from left arm with hyperkinetic movements are analyzed. The abscissa axis is time in seconds; the ordinate axis is frequency in Hz. Black arrow indicates the local maximum in the cross-wavelet spectrum.

**Figure 6 sensors-23-01531-f006:**
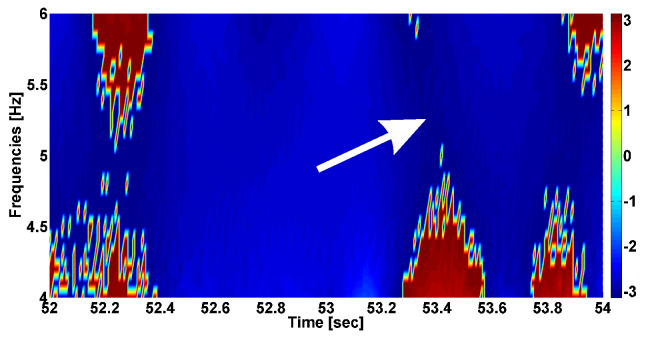
Phase shift of cross-wavelet spectrum given in Figure 5. The abscissa axis is time in seconds; the ordinate axis is frequency in Hz. White arrow indicates coordinates of the local maximum in the cross-wavelet spectrum.

**Figure 7 sensors-23-01531-f007:**
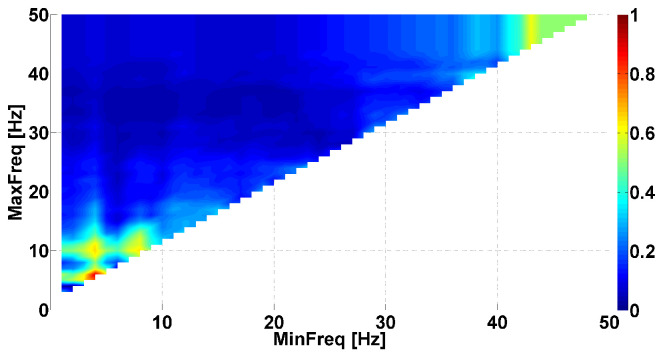
Example of Frequency cross-wavelet spectrum (CWS) AUC diagram. The tremor signals are analyzed in the antagonist muscles of the left arm of first-stage PD patients and ET patients. The abscissa axis is the lower bound of the frequency range; the ordinate axis is the upper bound of the frequency range.

**Figure 8 sensors-23-01531-f008:**
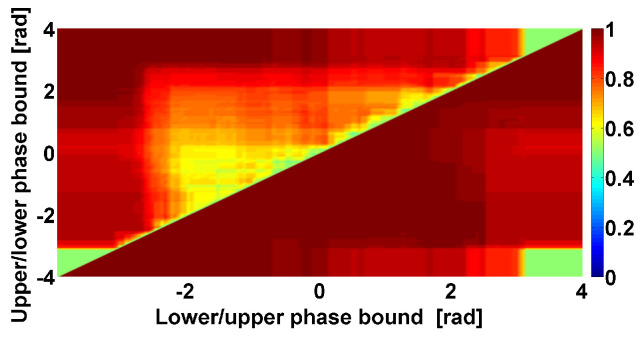
Example of Phase CWS AUC diagram. Tremor signals in antagonist muscles of left arm with hyperkinetic movements in PD and ET patients are analyzed. Cross-wave trains in the frequency range from 4 to 6 Hz are considered. In the upper triangular region: the abscissa axis indicates the lower bound of the phase range; the ordinate axis indicates the upper bound of the phase range. In the lower triangular region: the abscissa axis indicates the upper bound of the excluded phase range; the ordinate axis indicates the lower bound of the excluded phase range. The green rectangle in the lower left corner of the diagram is a construction artifact; it arose because the diagram axes were out of range from −π to +π. The same construction artifacts are the green triangles in the lower left and upper right corners of the diagram.

**Figure 9 sensors-23-01531-f009:**
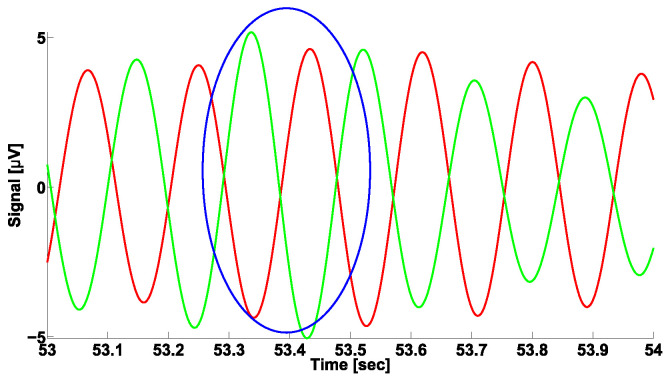
Example of the cross-wave train in frequency range from 4 to 6 Hz. Envelopes of EMG signals in antagonist muscles of left arm with hyperkinetic movements in a first-stage PD patient are given. The signals from the flexor and extensor muscles are indicated in green and red, respectively. Cross-wave train is indicated by a blue ellipse. The cross-wave train has the following characteristics: central frequency is 5.3 Hz, power spectral density (PSD) is 142.30 μV${}^{2}/$ Hz, duration is 1.39 periods, frequency bandwidth is 0.9 Hz, and instantaneous phase is −3.02 radians. The abscissa axis is the time in seconds. The ordinate axis is the envelope of the EMG signal in μV.

**Figure 10 sensors-23-01531-f010:**
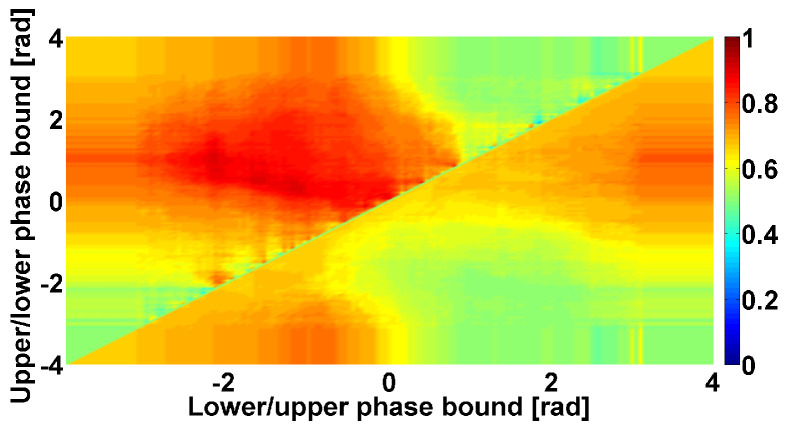
Example of Phase CWS AUC diagram. Tremor signals in antagonist muscles of left arm with hyperkinetic movements in PD and ET patients are analyzed. Cross-wave trains in the frequency range from 8 to 18 Hz are considered. The axes are the same as in Figure 8.

**Figure 11 sensors-23-01531-f011:**
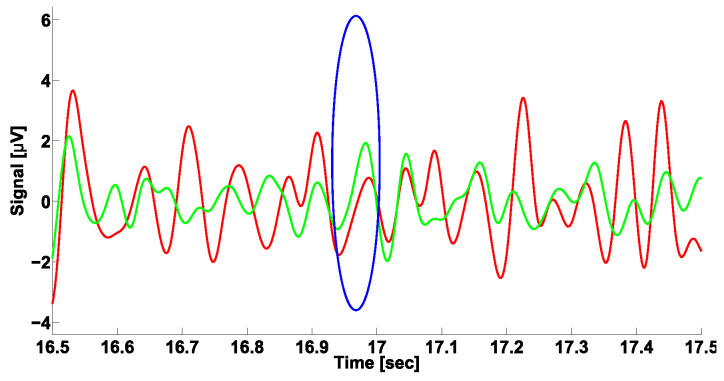
Example of the cross-wave train in frequency range from 8 to 18 Hz. Envelopes of EMG signals in antagonist muscles of left arm with hyperkinetic movements in a first-stage PD patient are demonstrated. The signals from the flexor and extensor muscles are indicated in green and red, respectively. Cross-wave train is indicated by a blue ellipse. The cross-wave train has the following characteristics: central frequency is 13.7 Hz, PSD is 3.86 μV${}^{2}/$ Hz, duration is 0.87 periods, frequency bandwidth is 2.6 Hz, and instantaneous phase is +0.3 radians. The abscissa axis indicates the time in seconds. The ordinate axis indicates the envelope of the signal in μV.

**Figure 12 sensors-23-01531-f012:**
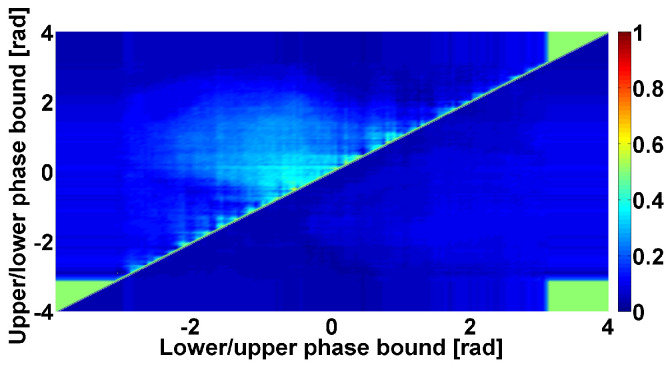
Example of Phase CWS AUC diagram. Tremor signals in antagonist muscles of left arm with hyperkinetic movements in PD and ET patients are analyzed. Cross-wave trains in the frequency range from 10 to 37 Hz are considered. The axes are the same as in Figure 8.

**Figure 13 sensors-23-01531-f013:**
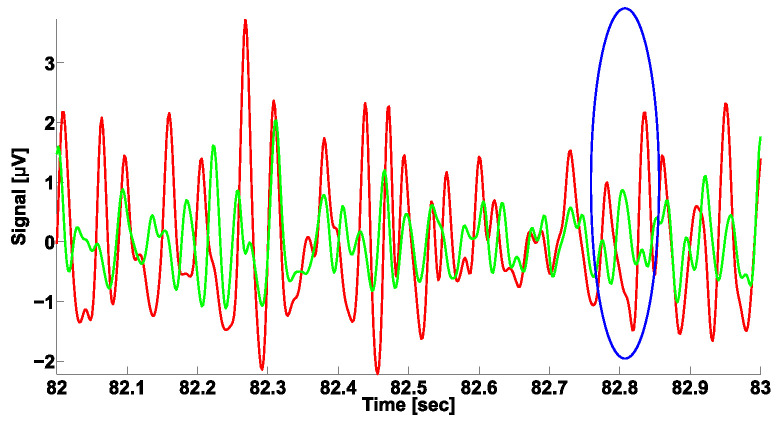
Example of the cross-wave train in frequency range from 10 to 37 Hz. Envelopes of EMG signals in antagonist muscles of left arm with hyperkinetic movements in ET patient are demonstrated. The signals from the flexor and extensor muscles are indicated in green and red, respectively. Cross-wave train is indicated by a blue ellipse and has the following characteristics: central frequency is 16.6 Hz, PSD is 0.69 μV${}^{2}/$ Hz, duration is 1.59 periods, frequency bandwidth is 4.3 Hz, and instantaneous phase is −2.14 radians. Abscissa axis indicates the time in seconds. Ordinate axis indicates the envelope of the signal in μV.

**Figure 14 sensors-23-01531-f014:**
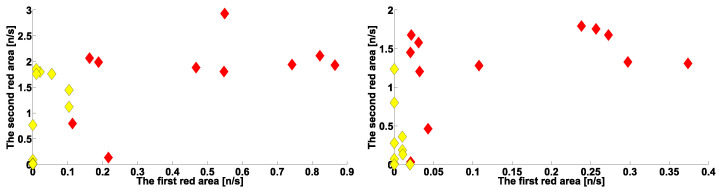
Scatter plot of the number of cross-wave trains per second in arms with hyperkinetic movements in PD and ET patients. Red diamonds are PD patients. Yellow diamonds are ET patients. (**Left**), the left arms in frequency ranges of 4–6 Hz (abscissa axis) and 8–18 Hz (ordinate axis). (**Right**), the right arms in frequency ranges of 2.5–6 Hz (abscissa axis) and 8–12 Hz (ordinate axis). Frequency ranges correspond to the rows in Table 1 and Table 2. On the coordinate axes, the symbols of the corresponding rows in the tables are indicated. They consist of the color of the plot area under consideration on the corresponding AUC diagram and the ordinal number of this area in the corresponding table.

**Figure 15 sensors-23-01531-f015:**
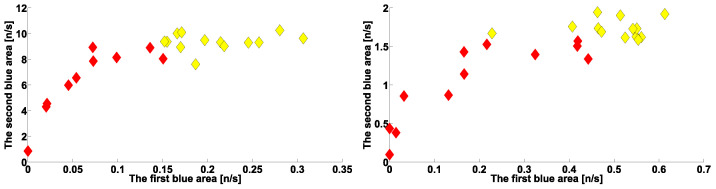
Scatter plots of number of cross-wave trains per second in arms with hyperkinetic movements in PD and ET patients. Red diamonds are PD patients. Yellow diamonds are ET patients. (**Left**), the left arms in frequency ranges of 2.2–3.9 Hz (abscissa axis) and 10–37 Hz (ordinate axis). (**Right**), the right arms in frequency ranges of 2.2–3.9 Hz (abscissa axis) and 12–37 Hz (ordinate axis).

**Figure 16 sensors-23-01531-f016:**
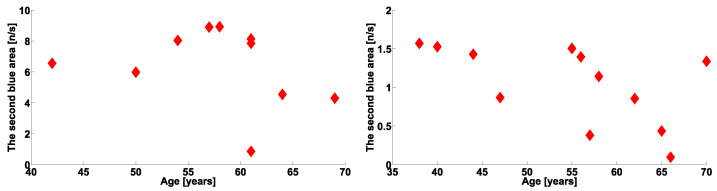
Scatter plots of number of cross-wave trains per second in arms with hyperkinetic movements in PD patients vs. patient age. The abscissa axis is the age in years. The ordinate axis is the number of cross-wave trains per second. (**Left**), the left arms of the subjects are considered: frequency range is 10–37 Hz. (**Right**), the right arms of the subjects are considered: frequency range is 12–37 Hz.

**Figure 17 sensors-23-01531-f017:**
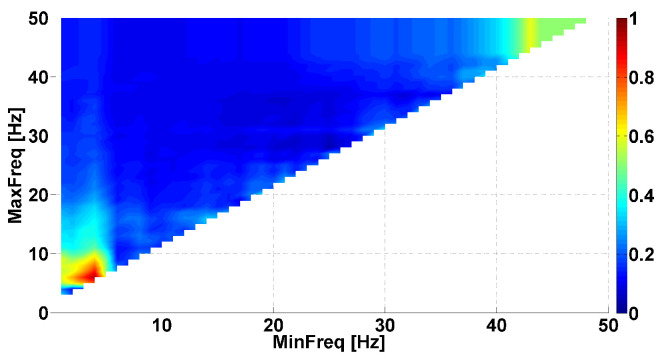
Example of Frequency CWS AUC diagram. The instantaneous phase of the considered cross-wave trains is constrained by the ranges from −π to −π/2 and from +π/2 to +π, that is, cross-wave trains related to the alternating tremor. Signals from the antagonist muscles in the *left* arm with hyperkinetic movement in PD and ET patients are considered. The abscissa axis is the lower bound of the frequency range; the ordinate axis is the upper bound of the frequency range.

**Figure 18 sensors-23-01531-f018:**
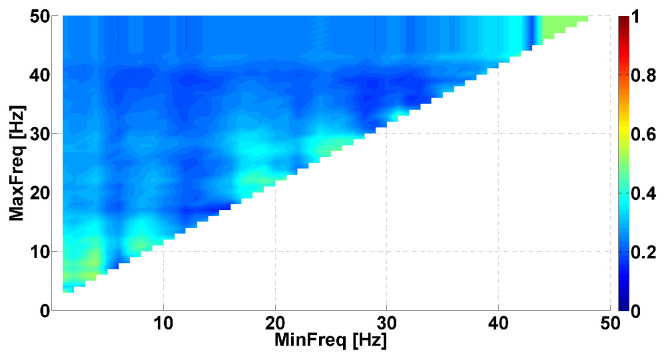
Example of Frequency CWS AUC diagram. The instantaneous phase of the considered cross-wave trains is constrained by the ranges from −π to −π/2 and from +π/2 to +π. Signals from the antagonist muscles in the *right* arm with hyperkinetic movement in PD and ET patients are considered. The abscissa axis is the lower bound of the frequency range; the ordinate axis is the upper bound of the frequency range.

**Table 1 sensors-23-01531-t001:** Characteristics of cross-wave trains that distinguish PD and ET patients with left arm hyperkinetic movements.

Investigated Regularity	Frequency, Hz	PSD, μV${}^{2}/$ Hz	Duration, Periods	Bandwidth, Hz	Phase, Rad	AUC	*p*-Value
The first blue area	2.2–3.9	0.01–27	0.6–1	0.7–1.4	−π…+π	0	0.00007
The first red area	4–6	2–4500	1–5	0.5–1.2	−π…+π	1	0.00005
The second red area	8–18	1–1200	0.4–2	1–8	−2.2…+1.1	0.89	0.002
The second blue area	10–37	0.02–65	0.2–2.4	1.5–22	−π…+π	0.03	0.0003

## Data Availability

The data presented in this study are available upon request from the corresponding author. The clinical data are not publicly available due to the ethical policy of the institute.

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
