# Peer review of "Investigation of Phase Shifts Using AUC Diagrams: Application to Differential Diagnosis of Parkinson’s Disease and Essential Tremor"

_sensors, 2023, doi:10.3390/s23031531_

Round 1

Reviewer 1 Report

The authors presented a study on phase shifts using AUC diagrams focused on the application to the differential diagnosis of Parkinson’s disease and essential tremor. Although the proposed topic would be a very useful contribution to the body of knowledge on this matter, I’m having concerns that need to be addressed by the authors was follows:

* Line 6: Authors assume that everyone reading the paper know the meaning of AUC, as it is not initially defined.

* Line 41: Please improve this sentence. Something like “Unfortunately, the relation between the pattern of tremor and neurodegenerative disease is ambiguous ”.

* Line 104: last sentence is unclear: “Note that the first….” I understand that during the first stage of the PD disease, PD patients develop hyperkinetic movements…” Am I right?

* Following my previous point, why is it any relevant dividing the patients based on the side of first hyperkinetic movements?

* Line 223: Authors claim to have found evidence of the “hypothesis that patients with hyperkinetic movements…” The issue here is that the mere comparison between graphs cannot be considered as sustainable evidence unless proper statistical comparisons are made. Similar argument applies for the entire paragraph starting in line 228. Perhaps, authors may be able to argue the validity of this visual comparison using new data mining techniques? Please, see below my comments on the Statistical analyses section of your paper.

* Line 309: same as above. It seems that only visual and descriptive comparisons are made among the AUC diagrams.

Overall: The conclusion section is clear and at the point, yet it does not correspond to the findings indicated in the paper. The starting sentence of the conclusion properly highlights all the working out that authors undertook: , “A new method of exploratory data analysis”. That is all. Please note I’m  not diminishing authors’ work whatsoever, as I found the research very compelling and a contribution to the literature in this regard. However, as pinpointed in all my previous observations, no statistical evidence of causal effects has been found. The statistical section of the paper incorrectly concludes a generalisation of the results using correlational analyses, which neither explain causality nor differences of the data studied. Thus, I would, for instance, rethink the sentence starting in line 525: authors’ results do not confirm the presence of “fundamental” differences...” But rather explored that those differences could potentially be generalised with further studies.

Author Response

Thank you very much for your comments on our manuscript. They certainly allowed us to improve our paper. Please, find our answers on your comments with indications of the introduced changes below. We hope that you will find our manuscript acceptable in its present form.
>>The authors presented a study on phase shifts using AUC diagrams focused on the >>application to the differential diagnosis of Parkinson’s disease and essential tremor. >>Although the proposed topic would be a very useful contribution to the body of >>knowledge on this matter, I’m having concerns that need to be addressed by the authors >>was follows:
>>* Line 6: Authors assume that everyone reading the paper know the meaning of AUC, >>as it is not initially defined.
Our answer: Thank you for the comment. Indeed, we wrongly supposed that all readers of Sensors know the AUC abbreviation. We added the explanation in the text. See lines 6 and 77.
>>* Line 41: Please improve this sentence. Something like “Unfortunately, the relation >>between the pattern of tremor and neurodegenerative disease is ambiguous”.
Our answer: Fixed. See line 44.
>>* Line 104: last sentence is unclear: “Note that the first….” I understand that during the >>first stage of the PD disease, PD patients develop hyperkinetic movements…” Am I >>right?
>>* Following my previous point, why is it any relevant dividing the patients based on the >>side of first hyperkinetic movements?
Our answer: Yes, you are right. During the first stage of PD, the patients develop hyperkinetic movements on only one side of the body. This fact is the main feature that differentiates the first stage of PD from next stages of the disease, when the second side also trembles. Additional explanations are included in the text. See line 108.
>>* Line 223: Authors claim to have found evidence of the “hypothesis that patients with >>hyperkinetic movements…” The issue here is that the mere comparison between graphs >>cannot be considered as sustainable evidence unless proper statistical comparisons are >>made.
>>Similar argument applies for the entire paragraph starting in line 228. Perhaps, authors >>may be able to argue the validity of this visual comparison using new data mining >>techniques? Please, see below my comments on the Statistical analyses section of your >>paper.
Our answer: Yes, thank you for pointing this out. Of course, the comparison of the graphs cannot be sustainable evidence. We can only make a hypothesis based on this observation. The text is now updated (see line 237). The statistical assessments of the observed regularities are added in the text using the new data mining technique. See lines 206, 217, 228, 235, and 339.
>>* Line 309: same as above. It seems that only visual and descriptive comparisons are >>made among the AUC diagrams.
Our answer: The Mann-Whitney test confirms that the number of cross-wave-trains differs significantly in PD and ET patients. This fact is confirmed for all kinds of cross-wave-trains reported in section 4. The statistical assessments of the observed regularities are added in the text (see Tables 1 and 2). Thank you for your idea to add statistical assessments in the paper.
>>Overall: The conclusion section is clear and at the point, yet it does not correspond to >>the findings indicated in the paper. The starting sentence of the conclusion properly >>highlights all the working out that authors undertook: , “A new method of exploratory >>data analysis”. That is all.
Our answer: We hope that the statistical assessments given in the paper make the conclusion section more substantial.
>>Please note I’m not diminishing authors’ work whatsoever, as I found the research very >>compelling and a contribution to the literature in this regard. However, as pinpointed in >>all my previous observations, no statistical evidence of causal effects has been found. >>The statistical section of the paper incorrectly concludes a generalisation of the results >>using correlational analyses, which neither explain causality nor differences of the data >>studied.
Our answer: Yes, you are right in that the presence of statistically significant correlation does not imply a causal relation between the correlated values. Thus, we can only make a hypothesis that may explain the observed correlations. We hope that the reader understands this issue.
>>Thus, I would, for instance, rethink the sentence starting in line 525: authors’ results do >>not confirm the presence of “fundamental” differences...” But rather explored that those >>differences could potentially be generalised with further studies.
Our answer: Fixed (see lines 542 and 544).

Reviewer 2 Report

   The abstract indicates the clear condition of examining differential phase for parkinson disease with some objective patterns. But the objective is not clear and if it not given as separate subsection then it is difficult to recognize. So for better reading Objectives must be included.

2.      In case of health applications what is the objective function and analytical model for parkinson disease?

3.      For all the defined advanced method the authors tried to prove it with experimental scenarios and they have integrated it optimization section. But no evaluation metrics for remaining scenarios are present. Therefore the outcome section can be extended.

4.      If the research is performed in real time then there is a need to provide some same reference model. In the research article the authors have compared the outcome with other data set which is quite interesting. But it is necessary to add another comparison with any one method with same data set. Comparisons with following papers can be made.

Hasanin, T.; Kshirsagar, P.R.; Manoharan, H.; Sengar, S.S.; Selvarajan, S.; Satapathy, S.C. Exploration of Despair Eccentricities Based on Scale Metrics with Feature Sampling Using a Deep Learning Algorithm. 2022.

To improve the quality of paper the authors must add some related articles. This can be in the previous 3-4 years where the same work has been carried out.

In the entire paper many grammatical mistakes are present. The authors must check the quality of writing.

Conclusion section must be enhanced with separate sub-section like advantages of proposed method in health industrial application etc.

Author Response

Thank you for your very careful review of our paper and for the comments, corrections, and suggestions. The revision of the paper has been carried out to take all of them into account. We believe that the paper has been significantly improved; although there is further research work left that we hope to address in a separate publication. Please, find our answers on your comments with indications of the introduced changes below.
>>The abstract indicates the clear condition of examining differential phase for parkinson >>disease with some objective patterns. But the objective is not clear and if it not given as >>separate subsection then it is difficult to recognize. So for better reading Objectives >>must be included.
Our answer: Thank you for this comment. The objective of this paper is development of a mathematical background of differential diagnosis of neurodegenerative diseases based on analysis of the phase difference of the biomedical signals. The objective is now added in the beginning of the Introduction section. See line 18.
>>2. In case of health applications what is the objective function and analytical model for >>parkinson disease?
Our answer: Please note that we do not use an analytical model for PD in this paper. We develop a mathematical method for analysis of the experimental data and revealing regularities that could be used for the differential diagnosis of PD and ET. The healthcare application of the results of the paper is a topic for a separate paper. Mathematical metrics can be developed that differentiate PD and ET patients during further research.
>>3. For all the defined advanced method the authors tried to prove it with experimental >>scenarios and they have integrated it optimization section. But no evaluation metrics for >>remaining scenarios are present. Therefore the outcome section can be extended.
Our answer: The statistical assessments of the observed regularities are added in the text (see lines 206, 217, 228, 235, 339, and Tables 1, 2). The Mann-Whitney test confirms that the number of cross-wave-trains differs significantly in PD and ET patients.
>>4. If the research is performed in real time then there is a need to provide some same >>reference model. In the research article the authors have compared the outcome with >>other data set which is quite interesting. But it is necessary to add another comparison >>with any one method with same data set. Comparisons with following papers can be >>made.
>>Hasanin, T.; Kshirsagar, P.R.; Manoharan, H.; Sengar, S.S.; Selvarajan, S.; Satapathy, >>S.C. Exploration of Despair Eccentricities Based on Scale Metrics with Feature >>Sampling Using a Deep Learning Algorithm. 2022.
Our answer: Thank you very much for this idea. We are interested in application of our method for the analysis of audio signals. Unfortunately, it looks like the pointed dataset includes no signals obtained in control group of patients. Please note that our method addresses the revealing differences between two datasets, for instance, PD and ET patients, or patients and healthy volunteers. In any case, this idea is a matter for further research and a separate paper.
>>To improve the quality of paper the authors must add some related articles. This can be >>in the previous 3-4 years where the same work has been carried out.
Our answer: Thank you for this comment. We have added several interesting research paper in the list.
>>In the entire paper many grammatical mistakes are present. The authors must check the >>quality of writing.
Our answer: Thank you for this note. We will make a professional proofreading of the text prior to the publication.
>>Conclusion section must be enhanced with separate sub-section like advantages of >>proposed method in health industrial application etc.
Our answer: Please note that our research is only the first step on the way to the industrial healthcare application. A lot of work is yet to be done in the future. Our further research will address development of mathematical metrics differentiating PD and ET patients that are necessary for the clinical applications of neurophysiological regularities reported in the paper. We have explained this point in the Conclusions section. See line 569.

Round 2

Reviewer 1 Report

I do appreciate authors taking the time to introduce reviewer's suggestions and fixes to be made.

Author Response

Thank you very much for your high appreciation of our paper. Following your recommendation, we have requested professional English proofreading.

Reviewer 2 Report

Is the research paper needs 80 references for comparing your work? What are all the basic needs for including such huge references? If such references are added then comparison statement must be made. In addition the comparison with the suggested paper is not made which indicates that given comments are not addressed by authors. 

Author Response

Thank you for your review. We understand your concern about the number of references needed for comparing our work. We agree with you that the number of references can be decreased. However, we suggested that the electronic format of the publication allows us to include all relevant papers that are interesting/significant from our point of view in the reference list. We are interested in experimenting with the acoustic data suggested by you. However, this must be separate research that takes sufficient time and requires interactions with the authors of the suggested paper and a detailed discussion of the acoustic data processing issues. Note that acoustic data is very different from EMG data. We agree with you that we observe frequency fluctuations both in the case of EMG and voice modulations. However, these types of fluctuations have a different neurophysiological and physiological basis. Thus, it is difficult to compare them. We have no experience in the investigation of voice modulations. This is a different area of neurophysiology and human physiology. This area is especially interesting for us because we know that some researchers investigate voice abnormalities in Parkinson’s disease patients. Please contact us directly if you are interested in such joint research. In accordance with your recommendation, we have requested professional English proofreading. We hope that this has addressed your concerns. Thank you once again for your review.